# Health sector readiness for the prevention and control of non-communicable diseases: A multi-method qualitative assessment in Nepal

**Bhim Prasad Sapkota**[1,2]*, **Kedar Prasad Baral**[3], **Ursula Berger**[4,5], **Klaus G. Parhofer**[6], **Eva A. Rehfuess**[4,5]

**1** CIH[LMU] Center for International Health, LMU Munich, Munich, Germany, **2** Teaching & Training Unit, Division of Infectious Diseases and Tropical Medicine, University Hospital, LMU Munich, Munich, Germany, **3** Patan Academy of Health Sciences, Lalitpur, Nepal, **4** Institute of Medical Information Processing Biometry and Epidemiology, LMU Munich, Munich, Germany, **5** Pettenkofer School of Public Health, Munich, Germany, **6** Medical Department-4, University Hospital, LMU Munich, Munich, Germany

* bhim.sapkota@lrz.uni-muenchen.de, bhimprasadsapkota@gmail.com

**Data Availability Statement:** All relevant data are within the paper and its Supporting information files.

## Abstract

In Nepal, deaths attributable to NCDs have increased in recent years. Although NCDs constitute a major public health problem, how best to address this has not received much attention. The objective of this study was to assess the readiness of the Nepalese health sector for the prevention and control of NCDs and their risk factors. The study followed a multi-method qualitative approach, using a review of policy documents, focus group discussions (FGDs), and in-depth interviews (IDIs) conducted between August and December 2020. The policy review was performed across four policy categories. FGDs were undertaken with different cadres of health workers and IDIs with policy makers, program managers and service providers. We performed content analysis using the WHO health system building blocks framework as the main categories. Policy documents were concerned with the growing NCD burden, but neglect the control of risk factors. FGDs and IDIs reveal significant perceived weaknesses in each of the six building blocks. According to study participants, existing services were focused on curative rather than preventive interventions. Poor retention of all health workers in rural locations, and of skilled health workers in urban locations led to the health workers across all levels being overburdened. Inadequate quantity and quality of health commodities for NCDs emerged as an important logistics issue. Monitoring and reporting for NCDs and their risk factors was found to be largely absent. Program decisions regarding NCDs did not use the available evidence. The limited budget dedicated to NCDs is being allocated to curative services. The engagement of non-health sectors with the prevention and control of NCDs remained largely neglected. There is a need to redirect health sector priorities towards NCD risk factors, notably to promote healthy diets and physical activity and to limit tobacco and alcohol consumption, at policy as well as community levels.

**Funding:** The author(s) received no specific funding for this work.

**Competing interests:** The authors have declared that no competing interests exist.

## Introduction

The global burden of non-communicable diseases (NCDs) constitutes a major public health challenge as well as a serious threat to social and economic development now and in the future [1]. The growing pandemic of NCDs is mainly caused by cardiovascular diseases, cancers, chronic respiratory diseases and diabetes [1, 2]. These four major killers contribute to more deaths globally than all other diseases combined [3]. The proportion of global deaths due to NCDs has increased from 68% in 2012 to 71% in 2016 [4]. NCDs account for 50% of all disabilities worldwide [5].

Low- and middle-income countries (LMICs) are being affected disproportionately, as they are home to around 80% of global NCD deaths and around 90% of early preventable NCD deaths [5]. NCDs as a share of total deaths are projected to increase by more than 50% in LMICs by 2030 [5]. Nepal has higher age-standardized death rates and disability-adjusted life years due to NCDs compared to communicable diseases [6]. The estimated contribution of NCD-attributable deaths has been increasing in Nepal: from 51% in 2010 and 60% in 2014 to 66% in 2016 [7]. Life expectancy at birth has been projected to increase from 67 years in 2011 to 71 years in 2019 [8], with the proportion of the elderly population (i.e. those aged 80 and older) increasing in parallel [8]. Aging is inevitable and unavoidable but the growing burden of NCDs among the elderly population is creating a challenge for "healthy aging" [9].

The major NCDs share common behavioral (i.e. tobacco, unhealthy diet, physical inactivity and the harmful use of alcohol) and biological risk factors (i.e. hypertension, hyperglycemia, hyperlipidemia and obesity) pointing to key pathways for prevention [10]. Several high-level commitments have recognized the preventability of NCDs and the urgent need for action to control them [11, 12]. The WHO Global NCD Action Plan with its nine targets for the prevention and control of NCDs and risk factors emphasizes national actions in the context of international cooperation and solidarity [1]. Based on cost effectiveness analysis, there are ten 'best buys' to address four major NCDs and their risk factors [13, 14].

Although NCDs constitute a major public health problem in Nepal, how best to address NCDs in primary health care and across different levels of the health system is not well described [4]. As a member state of the World Health Organization (WHO), Nepal is committed to achieving the targets of the 25x25 strategy of WHO, i.e. a relative reduction of premature deaths due to NCDs by 25% by 2025 [1]. As a UN member state, Nepal is also committed to achieving SDG target 3.4., i.e. a one third reduction of the premature mortality due to NCDs [15]. Nepal has initiated efforts to prevent and control NCDs through the ratification of a national policy in 2009, and through a strategy and plan of action in 2014 [16]. Readiness refers to the extent of willingness and ability of an organization to implement a particular intervention [17]. Health sector readiness is understood as the preparedness of health institutions to accept the challenges brought about by new health problems [18]. How the health sector has been preparing and responding to the growing pandemic of NCDs and its underlying risk factors has yet to be understood [19].

The aim of this study was to assess the readiness of the Nepalese health sector with regards to the prevention and control of NCDs and their risk factors by exploring the perspectives of policy makers, program managers and service providers. The study sought understand how NCD prevention and control in anchored in Nepalese policy documents and how different cadres of health workers as well as health policy makers and program managers at provincial and local levels perceive health sector readiness with regards to NCDs prevention and control.

## Materials and methods

### Study setting

Following the re-organization of the government health sector in the year 2015, the Nepalese health sector is characterized by three tiers of governance at federal, provincial and local levels. This study tried to capture insights across all three levels of the health system: document analysis was primarily concerned with the federal level, whereas qualitative research was carried out in three provinces–Bagmati, Gandaki and Karnali–which best represent the eco-political variations within Nepal. Within the three provinces, districts were purposively selected to cover the three distinct ecological regions of Nepal, i.e. mountains, hills and terai (plain). Kathmandu, Sindhupalchok and Chitwan district from Bagmati province, Kaski and Nawalpur districts from Gandaki province and Dailekh and Surkhet district from Karnali province were included in the study.

**Study design and participants.** The study adopted a multi-method qualitative approach, using a review of policy documents, focus group discussions (FGDs) and in-depth interviews (IDIs).

**Review of policy documents.** A wide range of policy decisions adopted by the Government of Nepal (GoN) were considered as "policy documents", notably the Nepalese constitution and various acts, regulations, policies, periodic plans, annual plans, strategies, guidelines, protocols and commission reports.

To identify policy documents, the official websites of the Ministry of Health and Population (MoHP, www.mohp.gov.np) and the Department of Health Services (DoHS, www.dohs.gov.np) were searched during the period August to September 2020. Any NCD- and NCD risk factor-related documents containing one of the following terms were considered for inclusion: 'policy', 'act', 'regulation', 'report', 'plan', 'program', 'strategy', 'guideline', 'protocol', 'package' and 'report'.

In addition, direct interactions with the policy section chief enabled access to a collection of policy documents at the MoHP. After de-duplication of records identified, a two-stage screening process was undertaken, initially screening for overall thematic relevance and subsequently screening for a matching of explicit terms related to 'NCDs' or their 'risk factors' in the contents (Fig 1).

Based on their title and/or contents, the shortlisted policy documents were categorised into four broad groups: legal policies, guiding policies, implementing policies and commission/study reports. Content analysis was performed within each category. To do so, the policy documents were read and reviewed throughly for the purpose of content familiarization. Subsequently, explicit statements mentioned in the policy documents about the prevention and control of NCDs and/or their risk factors were extracted onto a data extraction form. Qualitative information was organised using a deductive approach, sorting information according to the WHO health system building blocks framework, i.e. as health service delivery, health work force, health logistics, health financing, health information and surveillance, governance and leadership [20, 21].

**FGDs and IDIs.** FGDs were conducted with different cadres of health workers working in health posts (HPs), primary healthcare centers (PHCs), the municipality health section, municipal hospitals and provincial hospitals. IDIs were undertaken with policy makers, program managers and service providers in a complementary manner. FGD and IDI participants were selected purposively with a view to identifying multiple perspectives across different levels (i.e. provincial and local) and tasks of the health system and to ensure maximum variation. A semi-structured FGD guide and a semi-structured IDI guide were designed based on the six health system building blocks. Pilot testing for the FGD guide was conducted with health

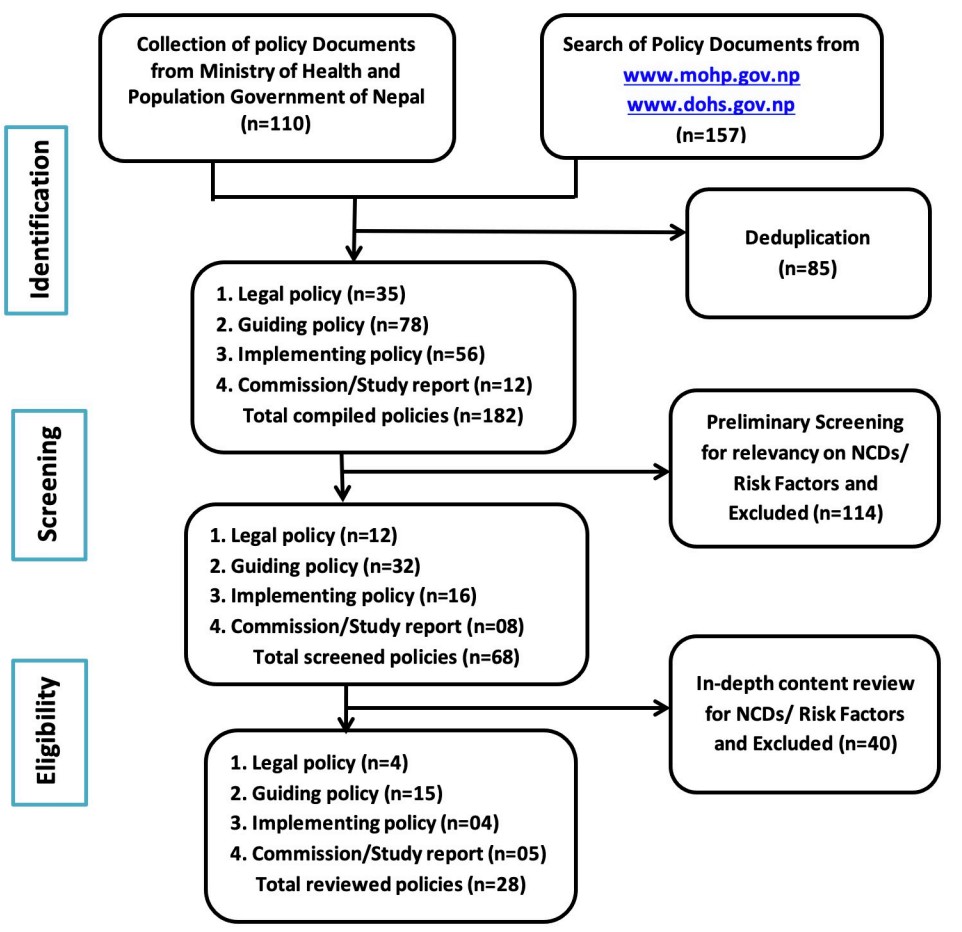

**Fig 1. Strategy for screening policy documents.**

workers at a primary healthcare center in Lalitpur. Pilot testing for the IDI guide was conducted with a mayor from Godavari municipality in Lalitpur. FGDs and IDIs took place between October and December 2020. They were moderated by an experienced researcher (BPS) and supported by a research assistant trained on how to conduct FGDs and IDIs who took notes. All discussions and interviews were audio-recorded. At the beginning participants were introduced to the objectives of the study, and written informed consent for their voluntary participation and audio recording was obtained. All FGDs and IDIs were conducted face-to-face in a convenient location at the participants' work place just after working hours. In doing so during the COVID-19 pandemic, appropriate public health measures were adopted and included physical distancing, the use of face mask and the use of sanitizer.

Audio recordings of the FGDs and IDIs were transcribed verbatim by the moderator (BPS) within 24 hours of the discussion or interview. The transcript was translated from Nepali to English by a bi-lingual expert and subsequently checked for accuracy of the translation by the moderator (BPS).

The transcripts were then analyzed manually using qualitative content analysis [22, 23]. The texts were reviewed iteratively and an initial set of codes was developed in an inductive manner. Codes were generated using the transcripts of the first two FGDs and IDIs; these were subsequently applied to the remaining FGDs and IDIs. From these codes themes were developed inductively and organized according to the six building blocks of the health system.

Initially, findings from the first FGDs and IDIs were compared with a view to examining differences between local and provincial levels; then, the same approach was applied to all FGDs and IDIs. In analyzing findings, we also specifically examined any reference made to the elderly population. Initial stages of the analysis (i.e. coding) were undertaken by a single researcher(BPS), subsequently, all authors engaged with the analysis and interpretation of data.

## Ethical approval

Ethical approval for the study was obtained from the ethical review board of the Nepal Health Research Council by reference letter no. 2882 dated 14 July 2020, and from the ethical committee of the Munich Medical Research School, LMU Munich, Germany by a letter dated 21.08.2020 (Hb/mbg, project no.20-657). Administrative permission to conduct FGDs and IDIs with government employees was obtained from the MoHP.

**Inclusivity in global research.**   The ethical, cultural and scientific considerations specific to inclusivity in global research has been fully maintained.

**Authors positionality.**   BPS is a male public health professional in Kathmandu, Nepal, and has spent almost two decades working in various functions in the Nepalese health sector. Both as a Ministry of Health employee and as a PhD student at the LMU Munich, where he further enhances his quantitative as well as qualitative research skills, he seeks to draw attention to the rising burden of non-communicable diseases in his home country, which may have influenced his interpretation of the data towards a need for action.

KPB is a male academic in Lalitpur, Nepal, having spent almost two decades in health system research and community health at national and sub-national levels.

UB is a female statistician and senior scientist at LMU Munich, new to qualitative research, but with long-standing expertise in quantitative research projects across a range of health topics, primarily in Europe but also in Africa and Asia.

KGP is a male professor of endocrinology and metabolism at the University hospital of LMU Munich, with a clinical and scientific focus on NCDs, notably obesity, diabetes and lipid disorders. His NCD-focused research has taken place in Germany, Europe and the USA, as well as in different Asian and African countries.

ER is a professor of public health and health services research at the LMU Munich, with a research focus on evidence-based public health and a strong command of a range of quantitative and qualitative research methods. A German female, she spent many years working for the World Health Organization and has been coordinating large research projects in sub-Saharan Africa; she has supervised several Nepalese PhD students, traveling the country and obtaining insights into how the Nepalese health system works. She firmly believes in the need to step up primary prevention and to address root causes of disease, a perspective that has sensitized her to focus on risk factors for disease.

## Findings

Of 182 unique policy documents identified, twenty-eight were included for content analysis, comprising four legal policies, 15 guiding policies, four implementing policies and five commission/study reports (Table 1).

We included 49 participants across eight FGDs and conducted 11 IDIs. None of the participants refused to participate and none dropped out. The average time taken for FGDs was 60 minutes (range: 50–70 minutes) while the average time taken for IDIs was 53 minutes (range: 45–90 minutes). Characteristics of the FGDs are shown in Table 2, characteristics of the IDIs are presented in Table 3.

**Table 1. List of policy documents.**

| Types of policy documents | Name of policy documents |
|---|---|
| **Legal policies** | 1. Constitution of Nepal 2016<br>2. Public Health Service Act-2018/Regulation 2019<br>3. Nepal Health Insurance Board Act 2015<br>4. Tobacco Products (control and regulatory) Act 2011/regulation-2012 |
| **Guiding policies** | 1. National Health Policy 2019<br>2. 15th Periodic Plan 2020–2024<br>3. 14th Periodic Plan 2017–2019<br>4. 13th periodic Plan 2014–2016<br>5. Nepal Health Sector Strategy 2015–2020<br>6. Integrated policy for prevention and control of NCDs 2006<br>7. NCDs Multi sectoral Action Plan 2014–2020<br>8. Urban Health Policy 2006<br>9. National Nutrition Policy and Strategy 2008<br>10. Multi sectoral Nutrition Plan 2018–2022<br>11. School Health and Nutrition Strategy 2006<br>12. National Population Policy 2015<br>13. National Policy on control and regulation of Alcohol 2017<br>14. Tobacco Control Strategy 2018<br>15. Nepal Health Sector Strategy Implementation Plan 2016–2021 |
| **Implementing policies** | 1. Guideline for Package of Essential NCDs training 2018<br>2. Guideline for the Poor Citizen Treatment Fund 2006<br>3. National Development Program 2020<br>4. Basic Health Service Package 2018 |
| **Commision/study reports** | 1. Assessment Report of NCD Multi-Sectoral Action Plan (2014–2020)-2019<br>2. The Nepal NCDI Poverty Commission National Report 2018<br>3. National Report on SDG Progress 2016–2030<br>4. Policy research on health-related SDGs in Nepal 2017<br>5. National Burden of Disease 2017 |

**Table 2. Characteristics of FGDs participants.**

| FGD No. | Categories of participants | Governing level | Province | Type of health facilities | Duration | Date | Number of participants | Eco-regional representation |
|---|---|---|---|---|---|---|---|---|
| FGD 1 | Health workers (Doctor, Paramedic, Nurses, Lab tech) | Local government | Bagmati province | Primary health center | 60 minutes | 9 October, 2020 | 7 (3 Male, 4 Female) | Rural mountain |
| FGD 2 | Health workers (Nurses, Paramedics, Lab technician) | Local government | Gandaki province | Municipal hospital | 50 minutes | 12 October, 2020 | 6 (3 Male, 3 Female) | Urban hill |
| FGD 3 | Doctors (Internist, Dental surgeon, Physician, Medical Officers) | Provincial government | Bagmati province | Provincial hospital | 55 minutes | 21 October, 2020 | 6 (4 Male, 2 Female) | Urban terai |
| FGD 4 | Health coordinators at municipality | Local government | Gandaki and Bagmati | Municipal level authorities | 70 minutes | 24 October, 2020 | 7 (7 Male, 0 Female) | Rural hill and terai (mixed group) |
| FGD 5 | Health posts in-charges in different HPs | Local government | Bagmati province | Local level health posts | 65 minutes | 28 October, 2020 | 5 (2 Male, 3 Female) | Urban hill |
| FGD 6 | Health woHP:(Nurses, Paramedics, Lab tech) | Local government | Bagmati province | Local level health post | 55 minutes | 3 November, 2020 | 5 (3 Male, 2 Female) | Urban terai |
| FGD 7 | Doctors (Medical officers, Dentist, Gynecologists, Medical Generalist) | Provincial government | Karnali province | Provincial hospital | 60 minutes | 20 December, 2020 | 7 (3 Male, 4 Female) | Urban mountain |
| FGD 8 | Health workers at health posts (Nurses, Paramedics, Lab tech) | Local government | Karnali province | Local level health post | 65 minutes | 23 December, 2020 | 6 (2 Male, 4 Female) | Rural hill |

**Table 3. Characteristics of IDIs participants.**

| No. of IDI | Governing level (province) | Interview setting/province | Type of responsibility | Duration of interview | Date | Gender of interviewe | Total experience (exp. of current position) |
|---|---|---|---|---|---|---|---|
| IDI No. 1 | Local Government (Bagmati) | Municipal office, Bagmati | Program manager | 45 Minutes | 9 October, 2020 | Male | 15 yrs. (1 yrs.) |
| IDI No. 2 | Provincial Government (Gandaki) | Ministry of Social Development, Gandaki | Policy maker | 60 Minutes | 12 October, 2020 | Male | 15 yrs. (1.5 yrs.) |
| IDI No.3 | Provincial Government (Gandaki) | Province Health Training Center, Gandaki | Program manager | 45 Minutes | 13 October, 2020 | Male | 15 yrs. (2yrs.) |
| IDI No.4 | Federal Government | Policy and Planning Section, MoHP | Policy maker | 45 Minutes | 16 October, 2020 | Male | 20 yrs. (2yrs.) |
| IDI No.5 | Province Government (Bagmati) | District Health Office, Bagmati | Program manager | 45 Minutes | 21 October, 2020 | Male | 8 yrs. (1 yr.) |
| IDI No. 6 | Local Government | Metropolitan city, Bagmati | Policy maker | 45 Minutes | 22 October, 2020 | Male | 15 yrs. (1 yrs.) |
| IDI No.7 | Local level (Gandaki) | Municipal Hospital, Gandaki | Service provider | 45 Minutes | 3 November, 2020 | Male | 4 yrs. (1 yrs.) |
| IDI No.8 | Province Government (Karnali) | District Health Office, Karnali | Program manager | 90 Minutes | 21 December, 2020 | Male | 5 yrs. (1 yrs.) |
| IDI No.9 | Province Government (Karnali) | Provincial Health Directorate, Karnali | Policy maker | 60 Minutes | 22 December,2020 | Male | 10 Yrs. (6 Months) |
| IDI No.10 | Local Government (Karnali) | Municipality Office, Karnali | Program manager | 60 Minutes | 23 December,2020 | Male | 10 yrs. (2yrs.) |
| IDI No.11 | Local Government (Karnali) | Health Post, Karnali | Service provider | 45 Minutes | 23 December,2020 | Male | 5 yrs. (2 yrs.) |

The insights gained from the review of policy documents and from the FGDs and IDIs are presented in an integrated manner for each of the six health system building blocks, exploring similarities and differences between provincial and local levels. The findings are reported as per the consolidated criteria for reporting qualitative research (COREQ) guidance [24].

## Health service delivery

Health service delivery was explicitly mentioned in the majority of policy documents, notably in three legal policies, four implementing policies, thirteen guiding policies, and four commission/study reports but this was focused on curative services. Preventive and health promotive services were mentioned only in a limited number of documents. The quality of health services regarding NCDs was suggested to be an additional problem.

"...despite the increased burden of NCDs due to globalization and life style change, there is a lack of quality and uniformity in health services..."

-15^th Periodic plan,2019–2024

Multi-sectoral engagement for NCD services with a view to tackling NCD risk factors was addressed in most of the policy documents. Notably, the National Health Policy 2015 specifies the need for integrated health services for the prevention and control of NCDs and their risk factors. Among the four behavioral risk factors of interest, interventions for tobacco control received the greatest policy attention, putting alcohol control in second place. Policy documents rarely referred to unhealthy foods or physical inactivity. Health services to meet the needs of the elderly population were not mentioned in any of the policy documents.

Most FGD participants and IDIs commented on the limited availability of NCD-specific health services, but some differences emerged with regards to the perception of these services at provincial and local levels. FGD participants at the local level emphasised that any available services for NCDs were part of basic health services. Similar opinions were found during an IDI with a provincial policy maker from Karnali province who had indicated the lack of NCD-specific curative services as well as services targeting NCDs risk factors. Municipal FGD participants highlighted that the package for essential NCD services (PEN) was available at the primary health care level. Provincial FGD participants reported that NCD services had been initiated in all three provinces; their effective implementation was only reported by FGD participants from Karnali province.

> "...there are no specific services targeting non-communicable diseases; medical consultation, free drugs, laboratory services and periodic health camps are a part of basic health services..."

> - FGD in Bagmati Province

A notable difference emerged for Bagmati province, where FGD participants described a community-based screening program targeting diabetes and hypertension. The program manager in Bagmati province also stressed the value of the initiatives of the school nurse program with regards to health promotion services. The program manager in Karnali province also provided some good examples of NCD-targeted services, e.g. cervical cancer screening and cash support for referral services. Health services targeting the health needs of the elderly population were not mentioned by any of the policy makers, program managers and health service providers across the three tiers of health system governance.

## Health workforce

Three guiding policies, two implementing policies and one commission/study report explicitly stressed the importance of the health workforce in the prevention and control of NCDs. These policy documents emphasize the training and skills of all cadres of health workers as well as the critical role played by community health workers for preventive and promotive interventions. They point to the critical scarcity of the health workforce in rural locations and the insufficient number of health workers trained on NCDs in urban locations. None of the policy documents identified the need for 'NCD service providers' as a new category of health workers. Despite the policy provision for the production of a service-oriented clinical workforce in the 15th periodic plan, none of the policy documents define quantitative targets for the training, recruitment, and retention of health workers designated for NCD-related health services.

> "...there is a lack of trained health workers for the care of elderly people suffering from chronic illness ..."

> -Policy research on health-related SDGs in Nepal,2017

Most of the FGD participants and IDIs mentioned that no health workers had been specifically designated for NCD-related services. Health workers at the local and provincial levels across all three provinces described that more than half of the allocated positions were vacant, even though the number of sanctioned positions compared to the daily workload is already very low. A similar opinion regarding the poor retention of trained health workers was expressed by municipal and provincial program managers in Bagmati province. All FGD participants mentioned that there was a crisis of NCD-specific health workers like ECG

technicians, dieticians, radiotherapy technicians, anesthesia technicians, operation theater technicians and other technical workers for NCD care. The municipal program manager in Bagmati province emphasized that the clinical knowledge and skills with regards to NCD treatment was poor among primary health care workers. In contrast, the program manager in Karnali province reported that there was no issue with retaining health workers; he stressed, however, that the poor competencies among them were an issue.

> "...despite designated positions, there has been no psychiatrist and cardiologist in provincial headquarters for the last six months,"
>
> -IDI in Gandaki province

Federal and provincial policymakers both mentioned that the domestic education and training of human resources for health (HRH) had improved in recent years but that the proportionate skill mix of health workers had not been fulfilled yet. The policymaker at the federal level indicated that health workers' sense of serving had been diverted to earning; as a result, trained health workers were migrating due to a poor retention strategy in Nepal.

> "...brain drain is one of the key human resources for health problems in the Nepalese health system, the retention strategy needs to be revisited..."
>
> -IDI in federal level

## Health logistics

Three guiding policies and one implementing policy described the importance of health logistics for the prevention and control of NCDs and their risk factors. Most of the policy statements on health logistics focus on drugs and few of them spell out the need for equipment/ instruments for diagnosis of NCDs. All policy documents mention that commonly used NCD drugs at primary health care level should be available to patients free of charge.

> "....NCD services aim to ensure the availability of basic technologies and essential medicines at the primary health care level free of cost....."
>
> - NCDs-Multi Sectoral Action Plan,2014–2020

Most FGD participants referred to the scarcity of NCD drugs, equipment and instruments at the primary health care level. They further mentioned that the regular supply of health commodities–as well as their quality–was a constant problem. FGD participants emphasized that most of the municipal authorities perceived drugs as 'the only commodity of relevance for NCD prevention and control, neglecting the need for diagnostic equipment and infrastructure. The problem of diverged costs and the poor quality of drugs and reagents procured by municipal authorities was raised during the IDIs with provincial policymakers in three provinces. Similarly, the interview with the federal policymaker revealed that local- and provincial-level purchase decisions were not guided by realistic demand forecasts, resulting in much variation in cost and quality.

> "....Procurement decisions are not guided by a procurement plan, and most of them have no idea about demand forecasting......"
>
> -IDI in federal level

The municipal program manager in Karnali province stated that NCD-related health commodities were sufficiently available. In contrast, health workers described that the supply of drugs and laboratory reagents from the municipality was neither regular nor timely.

*". . ..we send the demand form with the list of drugs required for a month but the supply is incomplete and sufficient only for half a month, the next supply is not certain. . .."*

*-IDI in Karnali Province*

Provincial FGD participants reported that the quantity of health commodities was sufficient but that regular and timely supply of commodities was not possible due to lengthy procurement processes. The program manager in Bagmati province stated that a modernized laboratory had been established at municipal level but that the quality of equipment, instruments, medicines and reagents was questionable due to the legal requirement to procure health commodities at the lowest possible cost. A notable comment by the provincial program manager in Karnali province revealed that medical equipment, once installed in hospitals, was rarely repaired. None of the policy makers, program managers and health service providers referred to the health logistics focusing on the health needs of the elderly population.

## Health financing

Two legal policies, five guiding policies, four implementing policies and three commission/ study reports described issues related to health financing for the prevention and control of NCDs. The review of these policy documents revealed the low priority given to NCD interventions during the allocation of financial resources. The policy statements also described that a lower than WHO-recommended proportion of the overall government budget was allocated to the health sector. None of the policy documents identified a dedicated budget for the prevention and control of NCDs, the available limited budget had been allocated to curative services. A report for the assessment of the NCD Multi sectoral Action Plan, 2014–2020 recommended that existing resources need to be re-directed towards preventive and promotive interventions targeting NCDs and their risk factors. The national report on SDG progress, 2016–2030 stated that existing priorities of external development partners and donors for NCDs were not sufficient. Some policy documents reported on a social security mechanism for poor and disadvantaged people but the administrative procedures of obtaining support were complex and the amount of support was insufficient relative to the cost of care. The national report of the Nepal NCDI poverty commission recommended that NCD interventions require 22% of total health expenditure or roughly $8.76 per capita expenditure or 1.4% of current GDP. Most of the policy documents referred to the progressive taxation on tobacco, alcohol and sugar-sweetened beverages as a key financing mechanism for tackling NCD risk factors.

*". . ... taxation on alcohol, tobacco and sugary drinks would be the financing technique for the prevention and control of NCDs and their risk factors. . . . . ."*

*- Integrated policy for prevention and control of NCDs, 2006*

Most of the FGD participants remarked that in ranking different sectors in terms of budget allocation, the health sector ranked below five. Furthermore, the budget allocated to NCDs was described as less than 10% of the overall health sector budget. The municipal program manager in Bagmati province believed the health sector budget to be sufficient but emphasized that the NCD-specific budget was negligible. Most of the IDIs from Gandaki and

Karnali provinces reported that the overall health budget was not satisfactory and that most of the budget allocated to NCD services was used to purchase drugs. Similarly, health workers in Karnali province mentioned that the NCD budget at municipal and provincial levels was assigned to drugs and diagnostics.

"……….politicians' priority is the construction of roads and bridges. Whatever budget is allocated for the health sector is spent on drugs for free distribution…….."

-IDI in Karnali Province

A notably different opinion was expressed by the provincial policy maker and program manager in Gandaki province. According to them, the provincial government paid appropriate attention to the financial needs of the health sector and prioritized NCDs over other health sector needs. Health workers in Bagmati province mentioned that the municipal health budget had been allocated to the free supply of drugs and laboratory tests targeting the elderly population.

## Health information and surveillance

Three guiding policies, two implementing policies and three commission/study reports referred to health information and surveillance. According to these, existing monitoring and reporting tools of the health management information system (HMIS) need to be revised to include more NCD-relevant variables; policy documents emphasized disease monitoring and tended to neglect risk factor monitoring. A report on policy research on the health-related SDG in Nepal explored the need for NCD-specific surveillance mechanism to ensure the effective and regular monitoring and assessment of NCDs and their risk factors. All policy documents mentioned the importance of using the evidence for sufficient drug supply, budget estimation and program decisions.

"…surveillance, monitoring, evaluation and research as one of the four action areas for realistic estimation of the burden due to NCDs…."

-NCD Multi sectoral Action Plan,2014–2020

All FGD participants mentioned that there was a well-established HMIS for recording and reporting of routine health services data but that monitoring and reporting mechanisms for NCDs and their risk factors was lacking. The municipal program manager in Bagmati province and the service provider in Gandaki province both expressed a similar opinion about the lack of NCD- specific surveillance mechanisms, notably with regards to risk factors. NCDs were rarely if ever discussed at monthly monitoring meetings in Bagmati and Gandaki provinces.

"…the municipal monthly review meeting for monitoring HMIS data focuses on priority public health programs, e.g. immunization, family planning, etc. NCDs and risk factors are never discussed…"

-IDI in Bagmati Province

Notably, the municipal program manager and service provider in Karnali province described the existence of a separate NCD monitoring and reporting mechanism in areas where the PEN package had been implemented. Here, NCD data were also discussed during the monthly monitoring meetings at municipal level. A service provider working in a health

post that implemented PEN reported that they were using a separate tool for monitoring and reporting on NCDs and their risk factors.

The federal level policy maker stated that data generated from the monitoring and reporting of routine health services had been used in the annual report but had not been used for program decisions.

> "….it is good to publish the annual report, but that report is being studied seldomly. Reviewing and using the evidence in annual reports for policy decisions should be promoted…"
>
> -IDI in federal level

## Governance and leadership

For the prevention and control of NCDs and their risk factors the issue of governance and leadership was explicitly mentioned in the majority of policy documents, notably in one legal policy, twelve guiding policies, three implementing policies and four commission/study reports. These revealed that some attention had been paid to the control of NCD risk factors through health promotion strategies, taxation of health-harming products and the construction of cycling lanes, pedestrian trails, public parks, gyms and yoga halls in urban area. These interventions require multisectoral engagement but, in reality, the contribution of sectors other than health is negligible.

> "… a need for multisectoral coordination with agricultural, environment and education sectors to enhance their role in health promotion…."
>
> -National Health Policy,2019

Municipal FGD participants mentioned that preventive and promotive services targeting NCDs were not prioritized. A similar opinion was expressed by municipal health coordinators in Gandaki and Bagmati provinces, where the health sector was not ranked among the top ten of fifteen priority sectors. IDIs with service managers revealed the perception of political leaders that spending on drugs and doctors was sufficient for achieving better health and that health investments beyond these aspects represented a waste of resources.

> "…for politicians plenty of free drugs and around-the-clock availability of doctors in hospitals are sufficient for better health of the people…."
>
> -FGD in Karnali Province

Provincial program managers in Bagmati, Karnali and Gandaki provinces stated that the overall priority assigned to the health sector by political leaders was very low because most of them had only limited awareness of the burden of NCDs and associated risk factors. In their view, this explained the higher priority given to hospitals, lab tests and drugs and the relative neglect of preventive and promotive services. A contradicting opinion was expressed by a provincial program manager in Karnali province who felt that political leaders and decision makers had sufficient knowledge about NCDs and their underlying risk factors.

> "…political leaders have good knowledge and ideas about NCD prevention and control, they draft very nice policy documents but their implementation is not so exciting…"
>
> -IDI in Karnali Province

All FGD participants and IDIs reflected that provincial governments paid greater attention to interventions targeting NCDs than local governments. Similarly, NCD-specific programs had received greater attention at the federal level compared to provincial and local levels. The federal policy maker reported that NCD-related services financed through federal budgets had been coordinated by the 'NCDs and mental health section' at the department of health services. He further stated that the 'poor consumption of the health budget' and 'program implementation focusing only on physical targets' are key governance issues. A critical concern was expressed with regards to the interrupted functional relationship between different levels of healthcare after the federal restructuring of the health sector.

"...we are focusing on the achievement of physical targets, not on the effectiveness and efficiency of health interventions......"

*-IDI in federal level*

## Discussion

This study is, to our knowledge, the first study that has examined the readiness of the Nepalese health sector with regards to the prevention and control of NCDs and their risk factors, based on the policy documents and, notably, the perceptions of the health workforce and health decision makers. Health sector policy documents identified in this study were found to be sensitive towards the growing burden of NCDs, but less so to the need to address risk factor for NCDs (Table 4).

Existing policy priority was thus focused on curative services with preventive and promotive interventions being neglected. Poor retention of trained health workers represents an old and continuous problem. More than half of the positions designated for NCDs were vacant resulting in too high a workload across all levels of health care. Inadequate quantity and quality of health commodities emerged as a key logistics issue. Inadequate budget allocation made for NCDs was also apparent, planning and program decisions regarding NCDs did not make use of available evidence. Engagement of non-health sectors with the prevention and control of NCDs was largely absent. In summary, a key finding obtained from this study is that the level of the health policy commitment is not sufficient to respond to the growing burden of NCDs. There is a gap between the design and implementation of policy into practice for the prevention and control of NCDs, with significant weaknesses apparent in each of the six-health system building blocks. Moreover, there is a need for a policy and program reform to redirect health sector priorities towards the control of NCD risk factors.

Many governments in LMICs seek a solution for the increasing health and financial implications of the NCD burden [25], however, the Nepalese effort appears to be motivated only by the health implications of NCDs. While the global status report on NCDs describes the vicious cycle between NCDs and poverty [26] none of the policy makers and program managers in Nepal consulted in this study made reference to these links.

In the UN Political Declaration on NCDs heads of state committed themselves to the development of national targets and indicators based on the specific national situation [7] Nepal has drafted a multisectoral action plan [16] in a timely manner, but national targets with relevance for the domestic context have not yet been set, as confirmed in this study. There is also a recommendation for routine and robust surveillance mechanism for NCD risk factors [27]. In Nepal, however, routine surveillance through the HMIS does not yet target NCD risk factors or NCD mortality, but instead focuses on NCD-related morbidity data.

**Table 4. Summary findings regarding perspectives on health sector readiness for NCD prevention and control.**

| SN | Building blocks | Key findings | |
| --- | --- | --- | --- |
| | | **Findings suggesting health sector readiness** | **Findings suggesting a need for improvement in health sector readiness** |
| 1 | Health service delivery | • NCDs services, where available, represent components of basic health services.<br>• The Package of Essential NCD services (PEN) is available at the primary care level.<br>• Services focused on curative services and targeting diseases are available. | • There are no specific health services targeting NCDs.<br>• Health services targeting the health needs of the elderly are not specified.<br>• Preventive and promotive services targeting risk factors receive far less priority than curative services. |
| 2 | Human resources for health | • PEN-trained health workers are more competent than non-PEN-trained health workers. | • In rural location there is poor retention of general health workforces but in urban location the trained health workforce is not available; reflecting brain drain.<br>• More than half of the positions are vacant, implying that those working are overburdened.<br>• There is no health workforce specially trained for NCD services. The required skill mix of the workforce is not being met. |
| 3 | Health commodities/ logistics | • Logistics focus on NCD drugs. | • Logistics rarely cover diagnostics, equipment, and instruments.<br>• Problems with health commodities include inadequate quality and quantity, and lack of a regular and timely supply.<br>• Repair and maintenance of equipment/instruments does not occur frequently. |
| 4 | Health financing | • During budget allocation, priority is given to curative aspects rather than preventive aspects. | • Limited priority is awarded to NCD-related activities in comparison with infectious disease-related activities.<br>• There is no dedicated budget for NCD-related interventions, and the budget consumption rate is slow.<br>• Funding support from external development partners for preventive interventions is negligible. |
| 5 | Health information and surveillance | • The current health management information system covers NCDs. | • The current health management information system does not cover NCD risk factors.<br>• Monthly monitoring meetings lack discussions about NCDs.<br>• NCD-related program decisions do not tend to use evidence. |
| 6 | Leadership and governance | • The 'NCDs and mental health section' at the MoHP coordinates NCD-related services.<br>• Some–but very few–efforts to address NCD risk factors have been made by local government, including the construction of cycle lanes, pedestrian trails, public parks, gyms, and a yoga hall. | • Interrupted functional relationships between different levels of healthcare after the restructuring of the health sector is a key governance problem.<br>• Provincial governments tend to pay greater attention to interventions targeting NCDs than local governments.<br>• Engagement of non-health sectors with prevention and health promotion activities is negligible; the health sector receives less priority compared to other sectors, at all levels of government and among political leaders. |

Scientific evidence demonstrates that the burden of NCDs and their associated risk factors can be reduced through effective preventive and health-promotive health services [10, 28]. However, health services offered at federal, provincial and local health facilities in Nepal concentrate on curative services. More broadly, existing wide disparities in access to health services and high out-of-pocket expenditures on health care are increasing in LMICs [29], with Nepal being no exception. The target for the allocation of government funding to health, which is recommended to be at least 22% of total expenditure [26], has not yet been achieved in Nepal.

Inadequate human resources in the health systems of LMICs represent a well-described problem [16]. The Nepalese health system suffers from, among others, a poor retention of

health workers and an inappropriate skill mix, with wide ranging impacts for population health in general and for NCDs in particular [30]. One approach to overcoming the latter problem would be to make training on the prevention, diagnosis and treatment of NCDs compulsory for all health workers, most importantly, a basic package needs to be incorporated in the preservice training curriculum of primary care health workers [31, 32].

Similarly, the availability of essential medicines and commodities, and functional logistics are critical components for primary health care to be able to respond to NCDs and their risk factors in an appropriate manner [32]. This study has shown that there is a problem with poor quality and quantity of essential medicines, laboratory reagents, equipment and instruments purchased and supplied by provincial and local governments in Nepal.

A study assessing service availability and readiness for NCDs in Bangladesh found that the NCD service available only target cardio-vascular diseases and diabetes mellitus and are only available at and above municipal level [33]. A qualitative assessment of health system preparedness in India [34] and Thailand [35] showed a low level of preparedness as well as low community engagement with NCD prevention and control. Low levels of readiness for major NCDs have also been commonly reported in Zambia [36] and Ghana [37]. All of these studies had used qualitative approaches with some quantitative triangulation.

The increasing burden of NCDs and an ageing population, means that efforts to prevent and control NCDs should pay particular attention to the growing elderly population in Nepal [38]. In this study, neither the documents reviewed nor the stakeholders interviewed explicitly referred to the specific health needs of Nepalese elders, neither in the context of NCDs nor with regards to meeting the targets under the UN decade of healthy ageing [39].

Age effects on NCDs and their risk factors are yet to be explored in the Nepalese context so that evidence-informed policies, plans and programs can be designed and implemented for the prevention and control of NCDs among older population.

## Strengths and weaknesses

A key strength of the present study is that it generated and combined insights from three distinct qualitative approaches (i.e. policy document review, FGDs and IDIs) and from a range of perspectives across different levels of the health system. It has also sought to capture insights from three provinces representing Nepal's three distinct eco-political regions. A weakness is that insights at the federal level were mostly drawn from policy documents and an interview with the single participant from the MoHP.

A significant limitation is that early stages of the analysis were only undertaken by a single person. Moreover, the study is solely based on qualitative methods albeit using different information sources. A mixed-methods study could contribute more representative quantitative insights and would enable the triangulation of qualitative and quantitative data.

Data collection for the policy documents made use of systematic searches; in addition, it benefited from the guidance of an MoHP official with almost two decades of experience in the Nepalese health sector. However, the review of policy documents focused on the health sector and may therefore have missed relevant policy documents drawn up by other sectors. We cannot exclude the possibility that BPS being the interviewer in FGDs and IDIS and working for the MoHP may have created social desirability bias among respondents.

Qualitative data analysis was guided by the WHO health system building blocks framework, thereby ensuring a structured approach and international comparability. However, qualitative data analysis was largely performed by one person (BPS) with input from all authors; joint coding and analysis of all data would have been preferable. Also, data were collected in Nepali, translated into English and then analyzed in English, which may have yielded minor errors in

translation and interpretation, although BPS checked all translations against the IDI and FGD transcriptions.

## Conclusion

This study has revealed important limitations in the readiness of the Nepalese health sector to respond the growing NCD pandemic, pointing to the need for reform. There is a critical gap in the formulation and implementation of NCD-targeted policies and community-based programs–notably with regards to implementing evidence-based interventions to promote healthy diets and physical activity and to strengthen preventive interventions concerned with tobacco and alcohol consumption–across the six building blocks of the health system.

With an increasing life expectancy, a growing elderly population will further exacerbate existing problems and this must be taken into account in advancing the health sector response to NCDs. Moreover, urbanization and changed lifestyles in Nepalese society must be considered, and sectors other than health involved with successful prevention and control of NCD risk factors.

Much of this study's assessment of health sector readiness is derived from the perceptions of different cadres of health workers and health decision-makers at national, provincial and community levels; future research should follow up with a quantitative assessment of health sector readiness, trying to triangulate qualitative and quantitative insights.

## Supporting information

**S1 Dataset. Minimum dataset used for qualitative analysis.**
(XLSX)

## Acknowledgments

We would like to thank Sagun Poudel, a research assistant, for note keeping during the FGDs and IDIs, and Shilu Subba, a bi-lingual translator, for translation of Nepali transcripts to English. We would like to acknowledge the Mayor of Godawari Municipality, Lalitpur for participating in the piloting of the IDI tool and health workers of Lubhu primary health center, Lalitpur for their participation in the piloting of FGD tool. We would also like to thank two anonymous peer reviewers for their thoughtful and constructive feedback that has strengthened the manuscript.

## Author Contributions

**Conceptualization:** Bhim Prasad Sapkota.

**Methodology:** Bhim Prasad Sapkota, Eva A. Rehfuess.

**Supervision:** Kedar Prasad Baral, Ursula Berger, Klaus G. Parhofer, Eva A. Rehfuess.

**Writing – original draft:** Bhim Prasad Sapkota.

**Writing – review & editing:** Bhim Prasad Sapkota, Ursula Berger, Klaus G. Parhofer, Eva A. Rehfuess.

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
