## [Decision Letter · Decision Letter 0]

13 May 2022

PONE-D-22-10244Health sector readiness for the prevention and control of non-communicable diseases: a multi-method qualitative assessment in NepalPLOS ONE

Dear Mr. Bhim Prasad Sapkota,

Thank you for submitting your manuscript to PLOS ONE. After careful consideration, we feel that it has merit but does not fully meet PLOS ONE’s publication criteria as it currently stands. Therefore, we invite you to submit a revised version of the manuscript that addresses the points raised during the review process.

We look forward to receiving your revised manuscript.

Kind regards,

Sebsibe Tadesse, PhD

Academic Editor

PLOS ONE

Journal Requirements:

We have noted that ethics approval from the ethical committee of the Munich Medical Research School was obtained after data collection was initiated. As PLOS ONE expected all human subject research to have ethics approval prior to data collection, we would be grateful if you could provide some clarification. 

Reviewers' comments:

**Comments to the Author**

1. Is the manuscript technically sound, and do the data support the conclusions?

Reviewer #1: Partly

Reviewer #2: Yes

2. Has the statistical analysis been performed appropriately and rigorously? 

Reviewer #1: N/A

Reviewer #2: N/A

3. Have the authors made all data underlying the findings in their manuscript fully available?

Reviewer #1: Yes

Reviewer #2: No

4. Is the manuscript presented in an intelligible fashion and written in standard English?

Reviewer #1: Yes

Reviewer #2: Yes

5. Review Comments to the Author

Reviewer #1: The topic is an important one no doubt and the approach to answer the research question is unique. However, the aim of the study needs to be re articulated to be coherent with the methodology used. There seems to be a mismatch- the aim is to assess readiness that is typically assessed by availability of infrastructure, equipment personnel etc. please see WHO tools for service availability and readiness (https://www.who.int/data/data-collection-tools/service-availability-and-readiness-assessment-(sara)?ua=1) but the methods are only qualitative.

2. The choice of a qualitative approach needs some justification- did the authors want to know why there are gaps in implementation or a lack of readiness? The choice of qualitative methods are usually to answer explanatory and exploratory questions - why and how questions..

3. Since the question is not well articulated the analysis also seems to be inconsistent

4. The analysis mentioned is sometimes as thematic, sometimes content analysis and also a framework is mentioned.

5. Thematic analysis should present in the results the main themes that emerged - headings from the framework can be used but themes need to be presented..what was the main finding is difficult to sense as a reader.

6. I would suggest reworking the question to frame a why is the situation the way it is kind of question- and then do a thematic analysis.

7. Please remove designations of persons interviewed in table 1 - I assume these would be fairly easy to identify and I find this unethical to mention in the paper. Knowing the designation is not critical to the analysis.

8. I have missed the codes or coding tree - please include this as an Annexure

9. Only 1 person doing the analysis is a limitation and should be mentioned as such. Another limitation is the lack of other approaches to data- quantitative data on personnel availability, facility assessments..triangulation of all sources would have added to the study.

10. There are discrepancies in the data- example regarding resource availability- how did the authors deal with the discrepancy- were follow up interviews done to understand the contradictions?

11. Please also state the authors positionally while conducting this research as this would be important to the interpretation of the findings.

Reviewer #2: The paper is a useful addition to the evidence on health sector readiness to deal with NCDs in South Asia. Some comments to strengthen the paper

Abstract: Needs to be shortened. Second line can be deleted. The building blocks should be in methods and not results. There is too much repetition of the fact that risk factors are not being addressed. The last line can also go.

Introduction can also be made crisper and shorter. In addition, to the section on NCD burden global and Nepal (one para) the next para has to be on global strategies for Prevention and Control of NCDs and their components (rather than listing declarations) and explain the concept of readiness.

Overall choice of study participants and the methods adopted are appropriate.

Results:

The use of health system building block framework is appropriate. Authors may want to revise the sequence in which the result is presented starting with governance & stewardship. Financing and ending with monitoring.

Currently one could link the statement of identifiable details of the respondents (place and position) and can be avoided.

Table 4 which shows the key results need to be organized better (retain the building blocks) to make it more informative. There are many ways that it can be done. For example, they could be split between policy review (can add the number of documents here rather than in text) component to FGD/IDI component. It could also be organized to show the good things and not so good things identified in the review.

Authors talk about NCD- specific worker and make this as a deficiency. However, more important and better approach is to include NCDs as a basic-package which is being done. This needs to be rethought.

Discussion should include some references to other country experiences and also implications for future.

6. PLOS authors have the option to publish the peer review history of their article (what does this mean?). If published, this will include your full peer review and any attached files.

Reviewer #1: No

Reviewer #2: **Yes: **Anand Krishnan

---

## [Author Response · Author response to Decision Letter 0]

5 Jul 2022

I have submitted the comprehensive response to all comments made by editor, and reviewer in tabular form.

---

## [Editor Report · Decision Letter 1]

19 Jul 2022

Health sector readiness for the prevention and control of non-communicable diseases: a multi-method qualitative assessment in Nepal

PONE-D-22-10244R1

Dear Dr. Bhim Prasad Sapkota,

We’re pleased to inform you that your manuscript has been judged scientifically suitable for publication and will be formally accepted for publication once it meets all outstanding technical requirements.

Kind regards,

Sebsibe Tadesse, PhD

Academic Editor

PLOS ONE

---

## [Editor Report · Acceptance letter]

27 Jul 2022

PONE-D-22-10244R1 

Health sector readiness for the prevention and control of non-communicable diseases: a multi-method qualitative assessment in Nepal 

Dear Dr. Sapkota:

I'm pleased to inform you that your manuscript has been deemed suitable for publication in PLOS ONE. Congratulations! Your manuscript is now with our production department. 

Kind regards, 

on behalf of

Dr. Sebsibe Tadesse 

Academic Editor

PLOS ONE